# The Effect of Alcohol on Cardiovascular Risk Factors: Is There New Information?

**DOI:** 10.3390/nu12040912

**Published:** 2020-03-27

**Authors:** Simona Minzer, Ricardo Arturo Losno, Rosa Casas

**Affiliations:** 1Department of Internal Medicine, Hospital Clinic, Institut d’Investigació Biomèdica August Pi iSunyer (IDIBAPS), University of Barcelona, Villarroel, 170, 08036 Barcelona, Spain; simona.minzer@gmail.com (S.M.); losnoric@gmail.com (R.A.L.); 2El Pino Hospital, Avenida Padre Hurtado 13560, San Bernardo, Santiago de Chile, Chile; 3CIBER 06/03: Fisiopatología de la Obesidad y la Nutrición, Instituto de Salud Carlos III, 28029 Madrid, Spain

**Keywords:** alcohol, cardiovascular health, inflammation, cardiovascular risk factors, hypertension, oxidative stress, dyslipidemia, type 2 diabetes

## Abstract

The effects of alcohol on cardiovascular health are heterogeneous and vary according to consumption dose and pattern. These effects have classically been described as having a J-shaped curve, in which low-to-moderate consumption is associated with less risk than lifetime abstention, and heavy drinkers show the highest risk. Nonetheless, the beneficial effects of alcohol have been questioned due to the difficulties in establishing a safe drinking threshold. This review focuses on the association between alcohol consumption and cardiovascular risk factors and the underlying mechanisms of damage, with review of the literature from the last 10 years.

## 1. Introduction

The effects of alcohol on health are various and heterogeneous and vary depending on the dose and pattern of consumption [1,2] (Figure 1). Heavy use of alcohol is one of the leading global risk factors for poor health outcomes, having a direct impact on a variety of diseases. It has been described that up to 19% of alcohol-attributable deaths were due to cardiovascular diseases (CVD) in 2016, after cancer and liver disease [1]. It is important to acknowledge the detrimental health effects of alcohol and its public health implications.

An alcoholic beverage has different definitions depending on the country and guideline revised. For example, in the United Kingdom an alcoholic beverage contains 8 g of ethanol, whereas in the United States and several European countries (i.e., Austria, France, Netherlands, Spain, among others) this value ascends to 10 g of ethanol [2,3]. The latter is the most frequently used measure, as stated by the World Health Organization (WHO) [3,4].

The effects of alcohol on health can vary depending on the amount (i.e., low-to-moderate versus heavy consumption) and pattern of intake (i.e., acute, binge, or chronic drinking) [5,6,7]. According to the Dietary Guidelines for Americans, the recommendations for moderate alcohol intake are considered as two standard drinks a day for men and one standard drink a day for women, which has been shown to reduce the risk of chronic disease. Moreover, according to the National Institute on Alcohol Abuse and Alcoholism (NIAAA) low-risk drinking is ≤3 drinks a day and <7 drinks per week for women, and 4 drinks a day or <14 drinks per week for men. Binge drinking is considered as 4 drinks for women and 5 drinks for men over a 2-hour period [2,7]. Nonetheless, in countries such as China, France, Ireland, and Spain, binge drinking is considered the consumption of 6 or more drinks on a single occasion [2].

Current drinking is defined as the intake of one or more drinks in the past 12 months [8]. In 2016, 32.5% of people worldwide were current drinkers, and the mean amount of alcohol consumed was 0.73 standard drinks daily for women and 1.7 for men [9]. The same year, the prevalence of current drinking was 72% and per capita alcohol consumption among adults in Europe (EU) was 11.3 liters of pure alcohol [1]. Nonetheless, consumption in the EU has decreased over the last few years, in contrast to consumption worldwide, which, in fact, is on the rise [1].

It should be noted that the effects produced by alcohol are not the same for all people. Its wide negative health effects are mainly attributable to differences in sex, inter-individual variability, type of alcoholic beverage consumed (fermented or distilled), amount (low, moderate), duration, patterns of intake (occasional, daily, binge), as well as socioeconomic factors [10,11,12,13]. On one hand, women present a greater sensitivity to the toxic effects of alcohol than men, showing decreased metabolism over the same amount of alcohol [12]. Genetics may play a role in inter-individual variability since it has been found that certain polymorphisms of the angiotensin-converting enzyme gene are associated with the development of alcoholic cardiomyopathy [12]. On the other hand, long-term heavy alcohol drinking is related to the development of diseases such as cardiovascular disorders (i.e., arrhythmias, cardiomyopathy, or stroke) or cancer (i.e., breast, esophagus, mouth, throat, colon, or rectum) [10]. In addition, excessive and chronic consumption of alcohol during pregnancy is associated with fetal alcohol syndrome, a wide spectrum of manifestations with severe implications. 

Moreover, excessive alcohol consumption is related to a higher risk of injuries and deaths by traffic accidents, suicide, marital violence, or child abuse, among others [12]. In contrast, light-to-moderate alcohol drinking has been linked with beneficial effect such as a reduction of risk of mortality by CVD, coronary heart disease (CHD), and stroke [12]. Socioeconomic status (SES) can also influence individual patterns of consumption [12,13]. For example, several studies have reported that people with a higher SES may consume similar or higher amounts of alcohol than those in the lower category. Nevertheless, groups with the lowest SES seem to present higher negative alcohol-related consequences [12]. Factors such as the level of education, race, ethnicity, and gender, as well as economic disparities and populations with marginalization and vulnerability can induce greater negative alcohol-related consequences [12,13]. For example, it seems that the higher the education level, the lower the presence of excessive drinking [12].

Behavioral risk factors such as overweight and obesity, poor diet, sedentarism or low physical activity, excessive alcohol consumption, and smoking have been correlated with a higher incidence and prevalence of CVD, diabetes, and dementia [8,14,15]. A recent systematic review reported that non-smoker individuals, with a moderate alcohol consumption, who were physically active and followed a healthy diet showed a lower risk of all-cause mortality (66%) than those that had none or only one of these healthy behaviors [15]. Ford et al. [8] reported similar results, showing that non-smoker individuals following a healthy diet and who were physically active presented a lower risk of mortality by CVD (65%) than those that have none of these healthy behaviors.

Accurate estimations of alcohol consumption can be challenging. The most frequent way to address this issue is by self-reporting, which inherently involves the possibility of under- or overestimating intake depending on the social perceptions of the individual [16,17,18]. Moreover, serum markers of alcohol intake have been identified, mainly gamma-glutamyltransferase (GGT). This marker correlates with alcohol consumption and has been shown to predict cardiovascular and all-cause mortality, independently of alcohol intake [18].

The importance the type of alcoholic beverage has on its health effects has been a subject of discussion. Studies examining the influence of beverage type have shown that there is no difference in CVD outcomes, and that the benefit comes from alcohol itself [6,18,19,20]. Thus, it appears that components other than ethanol could also exert beneficial effects [21]. It has been suggested that wine could provide greater protection because of its polyphenol content, but it is argued that lifestyle and time of day can play a more significant role in the wine-drinking population [5,18,19]. Nonetheless, evidence to support this theory is still lacking.

The cardiovascular health effects of alcohol have classically been described as having a J-shaped curve, in which low-to-moderate consumers present less risk than lifetime abstainers, and heavy drinkers show the highest risk [5,19,20,22,23,24,25]. Alcohol intake benefits not only healthy individuals, but also patients with established CVD [3,5]. This review will focus on the association between cardiovascular risk factors (i.e., hypertension, diabetes mellitus type 2, and dyslipidemia) and alcohol consumption and its underlying mechanisms of damage, with review of the literature from the last 10 years.

## 2. Search Strategy

In this article, a bibliographic review was carried out through PubMed, ScienceDirect, and Google Scholar from October 2019 to February 2020. This review is based on the most relevant articles and studies performed in human subjects published no longer than 10 years ago. The keywords used for this search were alcohol consumption, hypertension, diabetes mellitus type 2, dyslipidemia, inflammation, oxidative stress, among others.

The connectors employed were “AND” and “OR” combined with different keywords in order to find relevant articles for the work objective. Filters were used in the different databases to narrow the article search.

### Inclusion and Exclusion Criteria

Only articles that meet the following requirements were included:Articles published in the last 10 yearsWritten in Spanish or EnglishInterventions made in humansArticles published in journals with a relevant impact factor

The exclusion criteria consisted in the following:Articles published before 2010Articles not containing some of the characteristics mentioned in the inclusion criteriaInterventions made in animals, ex-vivo, or in-silicoArticles of meta-analysis or systemic reviews that may overlap with the studies mentioned in this review

## 3. Pathophysiology and Oxidative Stress

The factors responsible for the apparent cardiovascular benefits of light-to-moderate alcohol intake are uncertain. The inverse association between red wine consumption and mortality by CVD was initially published in 1979 [26]. Later on, in 1992, the concept of the “French Paradox” was introduced to describe an epidemiological observation in which the French show a relatively low incidence of CHD, despite the consumption of a diet rich in saturated fat and the presence of risk factors similar to those of other populations [27,28]. The relationship between alcohol consumption and cardiovascular events or all-cause mortality in apparently healthy people or patients with CVD has been depicted as a J-shaped curve attributed to a dose-related combination of beneficial and harmful effects [29,30].

### 3.1. Anti-Inflammatory and Antioxidant Effects

CVD are life-long, low-grade, chronic inflammatory, and oxidative diseases, initiated by elevated levels of low-density lipoprotein cholesterol (LDL-C) and its deposition in the intima of the blood vessels, forming atheromatous plaque. This plaque may eventually rupture, triggering the formation of a thrombus and leading to a major adverse cardiovascular event (MACE), defined as non-fatal myocardial infarction, revascularization by percutaneous coronary intervention or coronary artery bypass graft, stroke, and death by cardiac causes. In this setting, reduced concentrations of intermediate biomarkers of inflammation may reduce atheromatous plaque formation and decrease the risk of an event [31]. In a systematic review and meta-analysis, Brien et al. showed that moderate alcohol consumption seems to attenuate biological markers (increasing HDL cholesterol and adiponectin levels and decreasing fibrinogen concentrations) associated with the risk of CHD by modulating soluble markers [32].

Nova et al. published a cross-sectional study examining the association between moderate alcohol intake and the type of alcoholic beverage with inflammatory biomarkers. They evaluated 143 healthy individuals (>55 y) classified into three groups: abstainers and occasional consumers; beer consumers (beer ≥80% of total alcohol intake), and mixed beverage consumers. No differences were found between the consumption groups in parameters related to inflammatory markers, such as C-reactive protein (CRP) and cytokines. Further analyses showed that moderate wine, but not beer, consumption drove the association between alcohol intake and plasma high-density lipoprotein cholesterol (HDL-C), adiponectin, and soluble platelet (sP)-selectin levels [33].

The neutrophil-to-lymphocyte ratio (NLR) translates the balance between the detrimental effects of neutrophils and the benefits of active adaptive immune response and has prognostic implications in patient outcomes for multiple diseases. Howard et al. identified 48,023 adults who participated in the National Health and Nutrition Examination Survey (1999–2016). They described that abstinent participants (zero drinking days/year) and those who drank frequently (>100 drinking days/year) exhibited a higher NLR (2.06 and 2.01, respectively) than less frequent drinkers (1.95–1.96) [34].

As mentioned earlier, it has been suggested that the modulation of oxidative biomarkers could depend on the type of beverage consumed (Table 1). The evidence available at the time of a review by Covas et al. could not suggest that sustained wine consumption provided further antioxidant benefits in healthy individuals, but rather counteracted its own possible pro-oxidative effect. Nonetheless, the antioxidant effect of wine intake could be protective in oxidative stress situations [35]. Another study, published by Hamed et al. described that the consumption of red wine (RW) and two of its antioxidants, but not other sources of alcohol, prevents the activation of mononuclear cell nuclear factor kappa B (NF-KB), a redox-sensitive transcription factor involved in various processes that may contribute to atherosclerosis. Daily RW consumption for 21 consecutive days significantly enhanced vascular endothelial function in 20%. Although plasma stromal cell-derived factor-1 (SDF-1) concentrations remained unchanged, the endothelial progenitor cell (EPC) count and migration significantly increased after this period. RW increased the migration, proliferation, C-X-C chemokine receptor type 4 (CXCR-4) expression, and the activity of the Pi3K/Akt/eNOS signaling pathway and decreased the extent of apoptosis in glucose-stressed EPCs [36].

A German group evaluated dietary patterns and their association with inflammatory biomarkers. A sub-sample of 112 individuals from the Food Chain Plus (FoCus) cohort in Kiel were followed for a mean of 1.7 years. Amongst a wide set of variables measured; they described that beer intake was positively associated with inflammatory markers. It should be noted that all associations were not independent of measures of abdominal obesity [37]. Another cohort study published by Da Luz et al. followed 200 healthy male habitual RW drinkers (28.9 ± 15 g of alcohol/day for 23.4 ± 12.3 years) and compared them to 154 abstainers over a period of 5.5 years. RW drinkers showed significantly higher LDL-C and HDL-C levels, and significantly lower high-sensitivity CRP levels and MACE compared with abstainers. Contrary to a worse prognosis, calcification of the coronary arteries of RW drinkers might lead to plaque stabilization and fewer clinical events because of the polyphenols in RW [38].

Chiva-Blanch et al. studied the effects of the phenolic compounds of RW on inflammatory biomarkers of atherosclerosis. They developed a randomized, cross-over consumption trial in 67 high-cardiovascular-risk male participants, who received 30 g alcohol/d of RW, the equivalent amount of dealcoholized red wine (DRW), or 30 g alcohol/d of gin for 4 weeks. Interleukin-6 (IL-6) concentrations were significantly lower after the RW and DRW interventions compared with the gin intervention, whereas interleukin 10 (IL-10) concentrations were significantly higher after the RW and gin interventions compared with DRW. The authors suggested that both ethanol and the polyphenols in RW could be responsible for the modulation of inflammatory mediators in high-risk patients [39]. Another randomized controlled trial by Estruch et al. compared the effects of moderate RW and gin intake. They described that RW provided greater antioxidant effects, reducing plasma superoxide dismutase activity (7% reduction, *p* = 0.04) and malondialdehyde (9% reduction, *p* = 0.02) levels, probably due to its high polyphenolic content [40]. On the other hand, Stote et al. found no significant differences in the measurements of CRP and IL-6 concentrations in postmenopausal women after the consumption of 0, 15, and 30 g of alcohol per day for 8 weeks in a randomized cross-over design. Nonetheless, a decrease was observed in soluble intercellular adhesion molecule (sICAM) (5% for both doses), D-dimer (24% only for 30 g intake), and fibrinogen (4% and 6%, respectively) and an increase in plasminogen activator inhibitor type 1 (PAI-1) (27% and 54%, respectively) values, suggesting that moderate alcohol consumption (15–30 g/d) may provide beneficial effects on inflammation and hemostasis in postmenopausal women [41].

Recently in 2019, Roth et al. published a randomized, controlled, cross-over study in 38 high-risk male volunteers (55–80 y). The participants were randomized to receive 30 g of ethanol/day as aged white wine (AWW) or gin for 3 weeks. They described that following the AWW intervention, significant decreases in the concentrations of vascular cell adhesion molecule-1 (VCAM-1) (17%; *p* = 0.012), ICAM-1 (11%; *p* = 0.02), IL-8 (30%; *p* = 0.048), IL-18 (40%; *p* = 0.048), CD40, and CD31 T lymphocyte expression (*p* = 0.045) were observed compared with the gin intervention, while there were no changes in IL-1b, IL-6, and tumor necrosis factor alpha (TNF-a) concentrations. In addition, interferon gamma (IFN-g) concentrations significantly decreased after both AWW (33%) and gin intake (14%) (*p* = 0.046). There was also a significant 40% increase in EPC expression following AWW intake (*p* = 0.013), while gin had no effect. The authors concluded that AWW could maintain endothelial integrity with greater ability compared with gin, and that this ability could be attributed to grape-derived components [42].

### 3.2. Antithrombotic Effects

It has long been described that alcohol has important antithrombotic activity and, in this way, may achieve many of its cardiovascular benefits. In a 2010 literature review by Lippi et al. it was stated that the evidence available strongly supports a beneficial cardiovascular effect of low-to-moderate RW consumption (1–2 drinks/day; 10–30 g alcohol). This effect is mainly due to a decrease in oxidative stress, an increase in HDL-C, and a decrease in foam cell formation. It has also been associated with the upregulation of a number of fibrinolytic proteins, further favoring cardiovascular health [24]. Moreover, Lawlor et al. explored the causal effect of long-term alcohol consumption on CHD risk factors in 54,604 Danes (56 y). Amongst a number of measures, they found a weak nonsignificant inverse association between greater alcohol consumption and fibrinogen levels in a multivariable analysis (−2.0%, *p* = 0.32). Thus, the authors concluded that it is unlikely that the mechanism proposed to explain the lower CHD risk in moderate alcohol drinkers is produced by fibrinogen. Thus, there is a lack of robust evidence associating fibrinogen to CHD [39].

In a recent review, Wakabayashi investigated the relationship between platelet count and alcohol intake in 6508 middle-aged men who were either non-drinkers or drank less than 66 g/day. He described that there was no significant correlation between these two variables and concluded that further studies are needed to evaluate this association in heavier drinkers [40].

Smith et al. conducted a prospective, nonrandomized study in 54 healthy volunteers using thromboelastography with platelet mapping (TEG-PM) to evaluate platelet function after drinking either alcoholic or nonalcoholic drinks for a 2-hour period. In the alcohol consumption group, elastography (EG) in male subjects showed a higher median adenosin diphosphate (ADP) inhibition of platelet function compared with the non-alcohol group (15.7% (3.9, 39.3) vs. 8.2% (0, 30.1), *p* = 0.035), but this effect was not seen in the female participants. The authors concluded that there are differences in coagulation between genders, especially in resistance of the ADP receptor to alterations by acute alcohol intake [41]. 

In the same previously mentioned randomized, controlled study by Stote et al. including 53 postmenopausal women consuming increasing amounts of alcohol for 8 weeks, soluble intercellular adhesion molecule-1 (sICAM-1) concentrations decreased by 5% (*p* = 0.05) and D-dimer levels decreased 24% (*p* = 0.05) after the consumption of 30 g of alcohol. Fibrinogen concentrations significantly decreased by 4% and 6% and PAI-1 significantly increased 27% and 54% after the consumption of 15 and 30 g of alcohol, respectively. The authors concluded that in addition to improving the lipid profile, moderate alcohol consumption might also have beneficial effects on biomarkers of inflammation and hemostasis. Therefore, the lower risk for CVD linked to moderate alcohol intake in postmenopausal women might be explained by an additional, albeit to date unclear, mechanism [41].

Evidence suggests that moderate alcohol consumption—i.e., less than 30 g of alcohol per day-may have beneficial effects on inflammation, diminishing pro-inflammatory markers (e.g., IL-6, CRP) and raising anti-inflammatory markers (e.g., IL-10). The protective effect of alcoholic beverages could be related to the type of drink and the amount of alcohol ingested, as well as their polyphenol content [6,42].

## 4. Hypertension

Hypertension, defined as systolic blood pressure (SBP) over at least 140 mmHg and/or diastolic blood pressure (DBP) values of at least 90 mmHg, is a major risk factor for CVD [47,48,49,50]. Its overall prevalence in adults ranges between 30% and 45% [47]. According to the Global Burden of Disease Study, hypertension is one of the leading risk factors of early death and disability worldwide [50]. Many factors have been associated with the development of hypertension, such as an elevated body mass index, occupation, socioeconomic status, tobacco use, abdominal obesity, physical activity, and alcohol consumption, among others [5,48,51]. Many studies have reported a positive, dose- dependent association between alcohol intake and hypertension, showing the J-shaped curve also described for overall cardiovascular effects [5,16,19,22,52,53,54,55,56]. Excessive alcohol consumption accounts for about 16% of cases of hypertension worldwide [5,57].

Observational and prospective analytical studies have shown variable results (Table 2). A cross-sectional study of 6912 Polish men in the National Multi-center Health Survey (WOBASZ) showed a positive association between alcohol consumption and SBP and DBP. The authors concluded that moderate alcohol drinkers (15–30 g/d) had a 37% higher risk of hypertension while heavy drinkers (>30 g/d) had a 52% greater risk when compared with light drinkers (≤15 g/d) [55]. A cohort study following 2336 Japanese men for over 13 years described that the relationship between alcohol intake and CVD risk in hypertensive men without treatment was U-shaped, with the highest risk in never-drinkers [24]. A prospective cohort study of 32,389 Chinese men, with 4-year follow-up, described a dose-response relationship between daily alcohol consumption and the cumulative probability of hypertension. A positive, linear association was described between alcohol consumption and the risk of hypertension. Light-to-moderate alcohol consumption increased the risk of hypertension in men, and long-term alcohol intake at any dose was an independent risk factor for incident hypertension in this population [57].

Another cohort study following 1471 black South African individuals over a period of 5 years evaluated the association between alcohol intake, mortality, and the development of hypertension. They described that participants self-reporting alcohol intake had a 30% increased risk for developing hypertension (hazard ratio (HR) 1.30, 1.07–1.59) [16]. The same author published a sub-study of this cohort comparing self-reported estimates of alcohol intake with biochemical measures, describing that self-reporting of alcohol intake significantly predicted changes in blood pressure (BP), being the most accurate independent indicator for alcohol use in this population [17].

A dynamic prospective cohort of the Seguimiento Universidad de Navarra (SUN) study evaluated the relationship between the type, quantity, and frequency of drinking and the incidence of hypertension. After a 43,562 person-year follow-up, they found that individuals drinking >2 drinks/day had a higher risk of hypertension than non-drinkers (HR 1.55, 95% confidence interval (CI) 1.04–2.32). They described that a greater number of drinking days was weakly associated with a higher risk of hypertension. Finally, individuals who drank beer and spirits had a slightly higher, albeit not significant, risk of hypertension compared with wine drinkers (HR 1.18, 95% CI 0.97–1.44). This study suggested that the average quantity of alcohol consumption has a more important role in the risk of hypertension than the frequency of drinking [58].

Some studies have questioned the possible relationship between alcohol and the incidence of hypertension. A cross-sectional study with 12,285 individuals aged 37–66 reported that the daily consumption of alcohol beverages (from 0.1 to 15.0 g) was inversely related to the development of hypertension in women (OR 0.67, 95%CI 0.59–0.75, *p* < 0.001). In contrast, this relation was not observed for men [52]. Moreover, in a cohort study published in 2010, Halanych et al. described the relationship between five different categories of alcohol intake and the incidence of hypertension. They stated that alcohol use in general was not associated with the incidence of hypertension during a 20-year follow-up, except in European American women who presented a lower risk with any alcohol consumption [59].

The effect of the quantity of alcohol on hypertension according to the frequency of drinking is not yet clear, since studies remain contradictory. A study analyzing data of the 2005 US National Alcohol Survey, including 4083 individuals, evaluated the association between current drinking status and early life drinking patterns with three health conditions, including hypertension. They described an increased risk for hypertension in lifetime abstainers and a reduced risk in the less-heavy consumption group; nonetheless, when adjusted by a propensity score method, these findings were not significant. Current drinkers who drink 5 or more drinks/day at least monthly did show a significant risk of hypertension, meaning that consistent long-term heavy drinking is the real cause [60].

One issue that is important to consider when evaluating the contradictory results in prospective observational studies is the different populations included in the study. For instance, the association of alcohol consumption might be different in elderly patients compared with younger subjects. A cross-sectional study in a community-based elderly population of 553 high-cardiovascular-risk subjects with steady alcohol consumption over the previous 5 years evaluated the association between alcohol consumption and BP and its variability. It showed that moderate-to-heavy drinking is associated with higher BP values and that very light alcohol consumption was associated with lower daytime BP variability compared with no or occasional intake. Lower circadian BP variability may be beneficial for cardiovascular outcomes [61]. Jaubert et al. evaluated 533 older adults with a mean age of 70 ± 10 years. The mean ambulatory DBP was significantly higher in adults who consumed >1 drink/day, and SBP was not significantly different among study groups. Very light alcohol consumption (1 drink/month to 1 drink/week) was associated with reduced daytime BP variability [61].

It has also been reported that the effect of alcohol is different according to gender. Several studies have described that the J-shaped relationship between alcohol intake and hypertension could be sex-specific. Gender may be an important modifying factor of this benefit-risk relationship. A meta-analysis evaluating 20 cohort studies, including 361,254 participants, described that any alcohol consumption increased the risk for hypertension in men compared with abstainers. Yet, no difference in risk was observed in women who consumed 1 or 2 drinks daily compared with abstainers, but risk increased beyond this level. Thus, they concluded that there is an effect modification by sex and no protective association [53]. A second meta-analysis of 16 prospective long-term studies published in 2012 described that consumption of >20 g of ethanol per day significantly increased risk of hypertension in women, while in men it significantly increased with 31–40 g/day. Light-to-moderate consumption (<20 g/d) in women had a potential reduction of risk of hypertension, while men showed an increased risk. Moreover, the J-shaped curve relationship between alcohol intake and hypertension in women was found with a threshold of 10 g/day, but in men this relationship was more linear. Nonetheless, regardless of gender, all individuals consuming more than 20 g/day showed a higher risk of hypertension [56]. Another meta-analysis of 12 prospective studies described a linear, dose-response relationship between alcohol and hypertension in males, and a J-shaped curve for females, with protective effect at or below an intake of 15 g of pure alcohol/day, indicating that low average consumption may be protective for hypertension in females only. According to the authors, this could be explained by patterns of intake, since men show more binge drinking episodes, which may be linked to an increased incidence of hypertension [50].

Evidence from epidemiological studies has been corroborated by intervention studies in humans. For example, Mori et al. carried out a randomized controlled trial evaluating BP changes in 24 premenopausal women at three drinking levels (alcohol free, low volume, and high volume) during a 4-week period each. SBP and DBP were higher in women who consumed greater amounts of alcohol (2–3 drinks per day) compared with the other two drinking levels. However, lower amounts of intake did not show the BP-lowering effects evidenced in other studies [62]. A meta-analysis by Roereck et al. evaluating 36 clinical trials including 2865 participants described that a reduction in alcohol consumption reduced BP in a dose-dependent manner, when intake was over 2 drinks per day in both healthy participants and people with risk factors for CVD [49].

Alcohol has consistently been associated with the development of hypertension in various studies of different design. Nonetheless, its classical J-shaped curve has been questioned and associated only with women. For men, a dose–response effect seems more accurate, raising concern since they usually consume larger quantities of alcohol. Public health measures must be taken to reduce alcohol intake, specifically in high-cardiovascular-risk populations. These measures could have important and measurable effects, since alcohol-induced hypertension usually resolves in 1–4 weeks [5,18].

## 5. Dyslipidemia

Dyslipidemia is a group of diseases characterized by altered levels of blood lipoproteins and is a highly prevalent condition associated with cardiovascular events. A dose–response relationship has also been described between alcohol intake and blood lipids, especially with HDL-C, apolipoprotein-AI (apo-AI), LDL-C, and triglyceride (TG) levels [5,55]. It has been hypothesized that the possible mechanism by which alcohol promotes cardio-protective effects is by raising HDL-C levels [17,60,63,64]. This increase in HDL-C may be explained by the inhibitory effect of alcohol on the activity of cholesteryl-ester transfer protein (CEPT), which transports cholesterol from HDL to LDL particles, resulting in higher HDL concentrations [65]. On the other hand, heavy alcohol drinkers show higher levels of plasma TG than non-drinkers, due to an acute inhibition of lipoprotein lipase and improved synthesis of large very low density lipoprotein (VLDL) particles [64,66,67].

Observational studies have consistently reported that moderate alcohol intake is associated with lower CVD risk [63]. A cross-sectional study of 482 individuals from East China (Table 3) showed that daily alcohol intake was significantly associated with hypertriglyceridemia and that the duration of drinking was significantly associated with hypercholesterolemia [66]. Another cross-sectional survey conducted in China, evaluating 10,154 subjects, described that alcohol intake was positively associated with TG, HDL-C, and apolipoprotein A1 (apo-AI), and inversely associated with the LDL/HDL ratio and the apo-B/apo-AI ratio in males. This suggests that the beneficial effects of light-to-moderate alcohol consumption may be attributable to higher HDL or apo-AI levels and decreased lipoprotein-a (Lpa), and the harmful effects of heavy alcohol intake (< 30 g/d) may be attributable to increased TG and total cholesterol [68].

A cross-sectional study by Kwon et al. evaluated the relationship between drinking patterns and dyslipidemia using data from the 2010–2012 Korean National Health and Nutrition Examination Survey (*n* = 14,308). They described that high-risk drinking was associated with a higher risk for hypertriglyceridemia and elevated LDL-C in both sexes and was also inversely associated with hypo-HDL cholesterolemia [65]. Another study evaluated the effects of excessive alcohol intake at weekends in a young population of 180 male Spanish students. They observed that the group who consumed alcohol 2 days per weekend presented significantly higher levels of total cholesterol and TG compared with the non-drinker group. Moreover, the 1 day per weekend drinking group only showed higher TG levels compared with the nondrinking group [69].

A Korean study evaluated the relationship between alcohol intake and serum lipid levels in 1893 elderly men (> 60 y), extracting data from the 2005–2009 Korean National Health and Nutrition Examination Survey. They described that, in this population, alcohol intake was inversely associated with HDL-C concentrations and that TG concentrations increased with higher alcohol consumption [70]. Moreover, an additional cross-sectional multicenter study on 1896 Italian men (>65 y) investigated the association between different levels of current alcohol consumption and cardiovascular risk factors. They observed that elderly moderate drinkers tended to have higher levels of HDL-C and lower levels of systemic inflammatory markers and insulin resistance [71].

Prospective studies have also described a dose–response association between alcohol intake and the lipid profile. A cohort study following 6912 polish men showed a positive association between alcohol consumption and HDL and TG levels, noting that moderate drinkers had a 25% higher risk of elevated TG and a 40% lower risk of low HDL, while heavy consumption increased the risk of elevated TG by 46% and decreased the likelihood of low HDL by 44% [55]. Similar results were shown by a cohort study of 71,379 Chinese subjects followed over a 6-year period. Alcohol intake was associated with higher baseline HDL-C concentrations in a dose-response manner, and after follow-up they observed that the decreasing rate in HDL concentrations was lower with any amount of alcohol intake compared with never-drinkers and was lowest with moderate intake [63].

On the other hand, interventional studies have shown contradictory results. Prospective randomized studies have suggested that moderate amounts of alcohol can increase HDL-C [72]. The 2015 randomized controlled trial (RCT) by Mori et al. showed that higher levels of alcohol consumption were associated with a 10% increase in HDL-C [62]. Another trial described that moderate alcohol consumption (30 g/day) increased serum HDL-C by 5% and apo-AI by 6% [73]. A prospective, randomized, cross-over trial of 36 subjects showed that regular consumption of beer or alcohol-free beer in moderate quantities (2 cans/day for men and 1 can/day for women) significantly promoted protective properties of HDL-C, increasing its capacity to protect against LDL oxidation and enhancing cholesterol efflux [26].

On the contrary, a prospective randomized trial evaluated the effect of 90 days of moderate RW intake (150 mL/day for women, 300 mL/day for men) in 44 healthy subjects. They described no changes in either HDL-C or TG levels during the trial, yet LDL-C levels were lowered in the RW group (−0.3 mmol/L; 95%CI −0.6–−0.04). The reduction of LDL-C by RW was 16% when compared with the other groups at the end of the 90-day period [72].

The effect of alcoholic beverages on the lipid profile is attributed to their alcohol content. This was shown in a randomized, cross-over, clinical trial comparing the effects of moderate alcohol consumption (30 g/d) as gin or RW and their polyphenolic content and DRW on serum lipids of 73 male subjects at high risk of CVD. The mean adjusted LDL-C decreased 4.5% from baseline after RW and gin consumption, while HDL-C increased 7%from baseline during RW intake and 5% during gin intake compared with DRW. No significant changes were observed in total cholesterol and TGs [46].

A systematic review and meta-analysis evaluated 63 interventional studies (*n* = 1686 subjects) on the effects of alcohol consumption in biological markers of CHD. Pooled analysis showed a consistent increase in HDL-C in a significant dose-response manner, as well as apo-AI. However, alcohol intake did not significantly change total cholesterol, LDL-C, TG, or Lp(a) levels. Neither did pooled analysis stratified by dose show significant effects on LDL-C; nonetheless, there was a significant increase in TG at the highest dose of alcohol intake (>60 g/day) [32].

Several studies seem to support a dose–response relationship between alcohol and blood lipids, especially a reverse association for HDL-C, which may explain at least part of the cardio-protective effects of alcohol. Moreover, TG and total cholesterol levels can increase with heavy alcohol intake (>30 g/d). It is important to modulate alcohol consumption in order to obtain the beneficial effects of elevated HDL-C without potential detrimental cardiovascular effects

## 6. Type 2 Diabetes Mellitus

Diabetes Mellitus is a group of heterogenous diseases with variable clinical presentation and disease progression, in which hyperglycemia is a common characteristic [74]. Light-to-moderate alcohol consumption has been associated with a lower risk of Type 2 diabetes mellitus (T2D) in non-high-risk volunteers. 

Many prospective, analytical studies have examined the association between alcohol intake and the incidence of T2D (Table 4). Joosten et al. studied 35,625 adults of the Dutch European Prospective Investigation into Cancer and Nutrition (EPIC-NL) cohort, aged 20–70 years, who were free of diabetes, CVD, and cancer at baseline. After a follow-up of 360,661 person-years (median of 10.3 person-years) they verified 796 incident cases of T2D, who on comparison with abstainers showed HRs of 0.78 (95%CI 0.64–0.95), 0.55 (95%CI 0.43–0.69), and 0.57 (95%CI0.44–0.74) in light (0–4.9 g/d), moderate (5.0–14.9 g/d for women; 5.0–29.9 g/d for men), and heavy drinkers (≥15.0 g/d for women, ≥30 g/d for men), respectively. They did not find differences between the two genders. They further analyzed the risk of T2D incidence related to lifestyle patterns, concluding that moderate alcohol consumption reduces the risk of T2D in the presence of an individual low-risk lifestyle behavior (i.e., BMI, physical activity, smoking), but not because of a healthier lifestyle itself [75]. The same author studied 38,031 men, free of cancer or T2D, from the Health Professionals Follow-Up Study. During the 428,497 person-years follow-up, an increase in alcohol intake of 7.5 g per day was associated with a lower risk of T2D among initial non-drinkers (HR 0.78; 95%CI 0.6–1.0) and those initially consuming <15 g/day (HR 0.89; 95%CI 0.83–0.96). This observation was also described in light drinkers (0–4.9 g/day), with those increasing their intake to moderate levels (5.0–29.9 g/day) showing a significantly lower risk of T2D (HR 0.75; 95%CI 0.62–0.9). The authors concluded that an increase in alcohol consumption over a 4-year period was associated with a lower risk of T2D in rare and light drinkers [76].

The WOBASZ study evaluated 6912 men aged 20–74 years. Annual beer, wine, and vodka intake was assessed using a standardized questionnaire. After adjustment for confounding factors, they described that moderate drinkers (15–30 g/d) had a 35% lower risk of T2D compared with light drinkers (<15 g/d) [55]. Another cohort study published by Marques-Vidal followed 4765 participants without T2D at baseline for an average of 5.5 years. During follow-up, 284 participants (6%) developed T2D, and 643 (13%) developed impaired fasting glucose (IFG). Moderate alcohol consumption (14–27 units/week) tended to be associated with a lower risk of T2D but not for T2D + IFG. The multivariable-adjusted OR for T2D was 0.89 (95% CI 0.65–1.22), 0.66 (95% CI 0.42–1.03), and 1.63 (95% CI 0.93–2.84) for 1–13, 14–27, and 28 or more units per week, respectively (*p* < 0.005). This study showed no specific effect of alcoholic beverage type (i.e., wine, beer, or spirits) on T2D or IFG [77].

The effect of alcohol consumption on the incidence of T2D was also evaluated in the ATTICA study, a 10-year follow-up cohort, following 1514 men (18–89 y) and 1528 women (18–87 y). The average daily alcohol intake and type of alcoholic drink were assessed. The results showed that individuals who consumed up to 1 glass per day had a 53% lower T2D risk (relative risk (RR) 0.47; 95% CI 0.26–0.83) compared with abstainers, while trend analysis revealed a significant U-shaped relationship between the quantity of alcohol consumed and the incidence of T2D (*p* = 0.001 for trend). The protective effect of low alcohol consumption on the incidence of T2D was more prominent among individuals with stricter adherence to the Mediterranean diet (relative risk (RR) 0.08; 95% CI 0.011–0.70) and without metabolic syndrome (RR 0.34; 95% CI 0.16–0.70). Specific types of drinks were not associated with the incidence of T2D [78].

In recently diagnosed patients, small reductions in alcohol use are associated with a lower risk of CVD. A potential positive effect of moderate alcohol consumption, as manifested by higher HDL-C levels, could be counterbalanced by a negative effect on BP in high-risk patients [81]. Strelitz et al. followed a cohort of the Addition-Cambridge study of 852 adults in the year following diabetes diagnosis, and evaluated changes in diet, physical activity, and alcohol use. They described that a decrease of ≥2 units/week in alcohol intake was associated with lower risk of CVD (HR 0.56, 95% CI 0.36–0.87) [79].

Coronary artery calcification (CAC) is associated with atherosclerotic complications. However, elevated CAC may not always imply a worse prognosis. In 2018, Da Luz et al. described the clinical evolution of long-term RW drinkers in relation to CAC. They followed and compared 200 healthy male habitual RW drinkers with 154 abstainers for a period of 5.5 years. The RW drinkers ingested 28.9 ± 15 g of alcohol per day during 23.4 ± 12.3 years. The history of T2D was lower among drinkers, but other risk factors were similar. During the follow-up, MACE was significantly lower in drinkers than in abstainers, despite their higher CAC [38].

Finally, Baliunas et al. published a meta-analysis of 20 cohort studies evaluating the relationship between alcohol consumption and T2D. They described a U-shaped-relationship in both sexes; although compared with lifetime abstainers, the RR for T2D among men was most protective with an intake of 22 g/day (RR 0.87; 95% CI 0.76–1.00) and was deleterious over 60 g/day, while in women the consumption of 24 g/day showed the most protective effect (RR 0.60; 95% CI 0.52–0.69) and was deleterious over 50 g/day (RR 1.02; 95% CI 0.83–1.26). They concluded that moderate alcohol consumption is protective for T2D in men and women [80].

Multiple factors may play a role in T2D. The evidence reviewed suggests that light-to-moderate alcohol use is associated with a lower risk of T2D, describing a U-shaped relationship for men and women. No specific differences in alcoholic beverage type have been demonstrated. Nonetheless, reductions in alcohol use have been associated with lower risk of CVD, and therefore, recommendations should be adapted when T2D is diagnosed.

## 7. Conclusions

The effect of alcohol on cardiovascular risk factors was established long ago. This review shows that there are no definite associations, but trends can be specifically seen for each cardiovascular risk factor. Light-to-moderate alcohol intake, usually defined as <30 g/d, could be protective for hypertension and incident T2D and also shows an increase in HDL-C levels that could be cardio-protective. Future investigations must take into account the definition of levels and patterns of consumption, intake report by study subjects, measuring other characteristics and traits that can affect their response to alcohol consumption, and higher quality study designs are needed to avoid residual confounders. The deleterious effect of heavy alcohol consumption has been established by numerous studies, although the apparent beneficial effects of lower consumption are still under debate. Thus, recommendations must be made with caution since no definite conclusions have been made, and it is still a subject of study.

## Figures and Tables

**Figure 1 nutrients-12-00912-f001:**
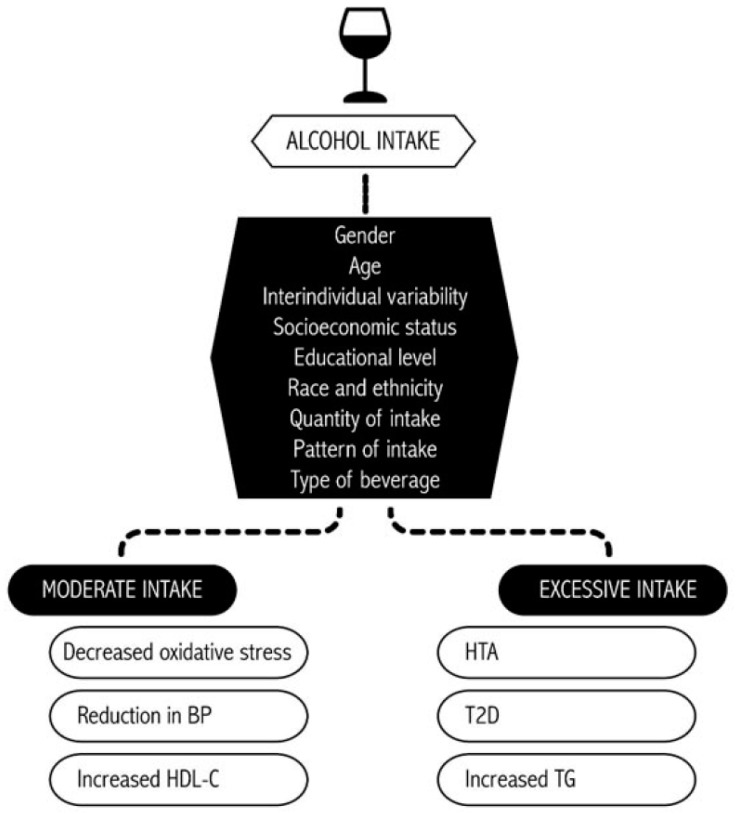
Alcohol’s health effects. BP, blood pressure; HDL-c, high-density lipoprotein cholesterol; HTA, hypertension; TG, triglycerides; T2D, type 2 diabetes.

**Table 1 nutrients-12-00912-t001:** Characteristics of studies evaluating anti-inflammatory, antioxidant, and antithrombotic effects of alcohol.

Reference	Design, Subjects (No), Follow-up	Population	Primary Outcome	Main Results
*Anti-inflammatory and antioxidant effects*				
**OBSERVATIONAL STUDIES**				
Nova et al. [33]	Observational, cross-sectional study, 143 participants. Participants were classified as abstainers and occasional consumers; predominantly beer consumers, and mixed beverage consumers (including wine, beer, and liquor).	Healthy adults 55 years of age and older. Spain.	Glucose, lipid profile, iron, transferrin, ferritin, and hs-CRP, VCAM-1, ICAM-1, P-selectin, E-selectin, IL-1β, IL-6, IL-8, IL-10, leptin, and adiponectin.	Consumption of alcohol: <25 g/d women and <40 g/d men. Mixed group: HDL-c and P-selectin compared with the abstainers’ group. Adiponectin compared with the beer group.
Howard et al. [34]	Cross-sectional, 48,023 participants. NHANES laboratory and questionnaire data from the nine surveys between 1999 and 2016.	All participants aged 18 years or older. USA.	Demographic, socioeconomic, and lifestyle factors associated with the magnitude of NLR.	NLR: Non-drinkers (zero drinking d/y): 2.06 Frequent drinkers (>100 drinking d/y): 2.01 Less frequent drinkers (<100 drinking d/y): 1.95–1.96
Hamed et al. [36]	Prospective, nonrandomized, and observational study, 14 participants consumed 250 mL red wine daily for 21 consecutive days.	Healthy volunteers. Israel.	Vascular endothelial function, plasma SDF1a concentrations, circulating EPCs and FC.	Moderate intake of RW (1–2 glasses/d) increased EPC migration and proliferation, NO production and decreased extent of apoptosis.
Barbaresko et al. [37]	Cohort study, 112 participants, mean follow-up of 1.7 years.	Northern German study population aged between 18 and 80 years.	CRP and IL-6 as response variables were used to derive dietary patterns.	Consumption of alcohol was associated with increase of CRP (OR 2.20; 95% CI 1.12, 4.35) and IL-6 (OR 3.14; 95% CI 1.26, 7.87).
Da Luz et al. [38]	Cohort study, 354 participants, follow-up of 5.5 years.	All males (200 healthy RW drinkers and 154 abstainers) aged 50–70 years. Brazil.	The composite endpoint of total death, AMI, or MACE was assessed.	RW drinkers (28.9 ± 15g/d) showed higher HDL and LDL but lower CRP than abstainers.
**INTERVENTIONAL STUDIES**				
Chiva-Blanch et al. [39]	Randomized, crossover consumption trial, 67 volunteers. After a washout period, the subjects received 30 g alcohol/d of RW, DRW, or gin for 4 wk.	High-risk, male volunteers between 55 and 75 y old recruited at Hospital Clinic of Barcelona.	Seven cellular and 18 serum inflammatory biomarkers were evaluated.	Moderate consumption of RW (30 g alcohol/d): Phenolic compounds modulate leukocyte adhesion molecules. Ethanol and polyphenols of RW modulate ICAM-1, IL-6, E-selectin
Estruch et al. [40]	Randomized cross-over trial. Forty participants received 30 g/ethanol/d as either wine or gin for 28 days.	Healthy men. Mean age, 38 years. Barcelona, Spain.	Serum vitamins, MDA, SOD, and glutathione peroxidase activities, lipid profile, oxidized LDL and LDL resistance to ex vivo oxidative stress.	RW (30 g/ethanol/d) intake: SOD activity (–8.1 U/gHb (95% confidence interval, CI, –138 to –25; *p* = 0.009)) and MDA levels (–11.9 nmol/L (CI, –21.4 to–2.5; *p* = 0.020)).
Stote et al. [41]	Randomized crossover design, 53 volunteers consumed a weight-maintaining diet plus 0, 15, and 30 g/day of alcohol for 8 weeks.	Women older than 50 years of age; postmenopausal (last menses ≥12 months before the beginning of the study).	s-ICAM, hs-CPR, IL-6 fibrinogen, PAI-1, D-dimer, Factor VII c, CRP, IL-6	After intake of 15 g and 30 g of alcohol, women showed decreased s-ICAM (5% and 5%), fibrinogen (4% and 6%), D-dimer (24% only for 30 g), and increased PAI-1 (27% and 54%). No difference for Factor VIIc, CRP, and IL-6.
Roth et al. [42]	Randomized, controlled, crossover study, 38 volunteers were randomized to receive 30 g of ethanol/day as AWW or gin for 3 weeks.	High-risk male volunteers between 55 and 80 y old recruited at primary care centers associated with Hospital Clinic of Barcelona.	Classical cardiovascular risk factors, cellular expression of circulating adhesion molecules, EPCs, and plasma biomarkers.	AWW (30 g ethanol/day) shows a greater ability to repair and maintain endothelial integrity compared with gin (39.6% increase in EPCs).
*Anti-thrombotic effects*				
**OBSERVATIONAL STUDIES**				
Lawlor et al. [43]	Causal associations in a cross-sectional study, 54,604 participants, Copenhagen General Population Study.	Danish General Population, mean age 56 years	BMI, BP, lipids, fibrinogen, and glucose	Non-drinkers were weakly associated with higher levels of fibrinogen.
Wakabayashi, [44]	Cross-sectional, 6508 participants.	Men aged 30–69 years. Japan.	BMI, Platelet count, leukocyte count, GGT	Mean platelet counts in drinkers, at any level of intake, were not significantly different from that in non-drinkers
Smith et al. [45]	Prospective, nonrandomized observational study, 54 volunteers.	Healthy volunteers, older than 21 years.	TEG-PM, maximal platelet activation, full platelet inhibition, ADP receptor agonist, or AA receptor agonist activation.	Acute alcohol consumption (maximum 2 g/kg) is associated with ADP receptor-mediated platelet inhibition in men, but not in women.
**INTERVENTIONAL STUDIES**				
Chiva-Blanch et al. [46]	Randomized, crossover consumption trial, 67 volunteers. After a washout period, the subjects received 30 g alcohol/d of RW, DRW, or gin for 4 weeks.	High-risk, male volunteers between 55 and 75 y old recruited at Hospital Clinic of Barcelona.	Fasting plasma glucose and insulin, HOMA-IR, plasma lipoproteins, apolipoproteins, and adipokines.	Beneficial effect of the polyphenols on insulin resistance (insulin and HOMA-IR for RW and DRW) with consumption of 30 g alcohol/d.

ADP: adenosin diphosphate; AWW, aged white wine; AMI, acute myocardial infarction; Apo-AI, apolipoprotein; BMI, body mass index; CHD, coronary heart disease; CI: confidence interval; CPR, C-reactive protein; CV, cardiovascular; DRW, dealcoholized red wine; EPC, endothelial progenitor cell; FC, flow cytometry; GGT, gamma glutamyl transpeptidase; HDL-C, high-density lipoprotein cholesterol; HOMA-IR, homeostasis model assessment of insulin resistance; HTG, hypertriglyceridemia; ICAM-1, intercellular adhesion molecule 1; IL, interleukin; LDL-C, low-density lipoprotein cholesterol; MACE, major adverse cardiovascular events; MDA, malondialdehyde; NLR, neutrophil-to-lymphocyte ratio; PAI-1, Plasminogen activator inhibitor-1; RW, red wine; SOD, superoxide dismutase; TC, total cholesterol; TEG-PM, Thromboelastography with platelet mapping; TG, triglycerides; VCAM-1, vascular cell adhesion molecule 1. SDF1a, stromal cell-derived factor-1.

**Table 2 nutrients-12-00912-t002:** Characteristics of 14 studies evaluating the effect of alcohol on hypertension.

Reference	Design, Subjects (No), Follow-up	Population	Primary Outcome	Main Results
**OBSERVATIONAL STUDIES**				
Zatu et al. [16]	Cohort study, *N* = 1471, 5 years	PURE study, Black South Africans	HT incidence, mortality	Self-reported intake is associated with 30% higher risk of HT
Higashiyama et al. [24]	Cohort study, *N* = 2336, 13 years	Suita study, urban Japanese men	HT and CVD risk	U-shaped risk in hypertensive subjects without treatment
Suliga et al. [52]	Cross-sectional, *N* = 12,285, aged 37–66 years.	Polish-Norwegian Study (PONS) project	Prevalence of CVD	Both in men and in women, alcohol consumption was related to a lower prevalence of CVD.
Waśkiewicz et al., [55]	Cross-sectional, *N* = 6912	National multi-center health survey (WOBASZ), polish men	CV risk factor profile	Moderate drinkers (15–30g/d) have 37% and heavy drinkers (>30 g/d) 52% greater risk of HT
Peng et al. [57]	Cohort study, *N* = 32,389, 4 years	Kailuan study, Chinese male coal mine workers	HT incidence	Long-term intake is independent risk factor at any dose
Núñez-Córdoba et al. [58]	Cohort study (dynamic), *N* = 43,562 person-year follow-up	SUN study, Spanish population	HT incidence	>2 drinks/day has HR 1.55 for HT than non-drinkers
Halanych et al. [59]	Cohort study, *N* = 4711, 20 years	Development of coronary artery risk in young adults study cohort, United States	HT incidence	Alcohol use not associated with HT incidence
Kerr et al. [60]	Cross-sectional, *N* = 4083	US National Alcohol Survey 2005	HT risk	≥5 drinks/day at least monthly has higher risk of HT
Jaubert et al. [61]	Cross-sectional *N* = 553	Elderly population with high CV risk and steady alcohol consumption	BP variability	Intake >1 drink/day shows higher BP values
Mori et al. [62]	Randomized-controlled study, *N* = 24	Healthy premenopausal women	Changes in BP according to drinking level	Higher BP levels with 2–3 drinks/day
**META-ANALYSES**				
Roerecke et al. [49]	Meta-analysis of 36 clinical trials, *N* = 2865	General population	Effect of reduction in alcohol consumption in BP	Reduction in alcohol consumption lowers BP in dose–response manner
Taylor et al. [50]	Meta-analysis of 12 cohort studies, *N* = 27,603	General population, United States, Japan, Korea	HT risk	Dose-response relationship in males, J-shaped curve in females (>15 g/day)
Roerecke et al. [53]	Meta-analysis of 20 cohort studies, *N* = 361,254	General population	HT incidence	Higher risk for men with any level of consumption, higher risk in women consuming >2 drinks/day
Briasoulis et al. [56]	Meta-analysis of 16 cohort studies, *N* = 227,656	General population	HT incidence	>20 g/day has higher risk of HT, J-shaped curve in women, linear relationship in men

BP, blood pressure, CV, cardiovascular, CVD, cardiovascular disease; HT, hypertension.

**Table 3 nutrients-12-00912-t003:** Characteristics of 14 studies evaluating the effect of alcohol on dyslipidemia.

Reference	Design, Subjects (No), Follow-up	Population	Primary Outcome	Main Results
**OBSERVATIONAL STUDIES**				
Waśkiewicz et al. [55]	Cross-sectional, *N* = 6912	National Multicenter Health Survey (WOBASZ), male population, Poland. Aged 20–74 years	Dyslipidemia	Intake >30 g/d increased risk of HTG by 46% and decreased likelihood of low HDL-C by 44%. Daily intake is associated with HTG.
Huang et al. [63]	Cohort study, *N* = 71,379, 6 years	General population China	Change in HDL-C	Umbrella-shaped association, moderate intake (0.5–2 serving/d) showed lowest rate of HDL-C.
Kwon et al. [65]	Cross-sectional, *N* = 14,308	Korean National Health and Nutrition Examination Survey, 2010–2012	Dyslipidemia	Low-risk drinkers, 0–7; intermediate-risk drinkers, 8–14; high-risk drinkers, ≥ 15 points (using AUDIT score). High-risk drinking is associated with HTG and elevated LDL-C in both sexes, and inversely associated with lower HDL-C.
Shen et al. [66]	Cross-sectional, *N* = 482	Age- and sex-matched individuals from East China	Dyslipidemia	Drinker is defined ≥25 g of alcohol/d. Age and BMI were associated with hypercholesterolemia. Daily alcohol intake was associated with hypertriglyceridemia.
Hao et al. [68]	Cross-sectional, *N* = 10,154	General population, East China Urban and rural population, China	Serum lipids	Higher TC, HDL-C, and Apo-a1 with increase in alcohol intake (Heavy alcohol drinkers, ≥30 g/d).
Timón et al. [69]	Cross-sectional, *N* = 180	Young university students (18–22 y), Spain	Effects of excessive alcohol drinking CV risk factors	Drinking 2 days/weekend shows higher TC and TG levels.
Park et al. [70]	Cross-sectional, *N* = 1893	Elderly men (>60 y) from the Korean National Health and Nutrition Examination Survey, 2005–2009	Serum lipids	None, very light (0.1–4.9 g/day), light (5.0–14.9 g/day), moderate (15.0–29.9 g/day), and heavy (≥30 g/day). Alcohol consumption was negatively associated with a risk for ↓ HDL and ↑ TG levels (*p* for trend < 0.001; both).
Perissinotto et al. [71]	Cross-sectional, *N* = 1896	Elderly men (>65 y), Italy	Cardiovascular risk factors	Rising linear trend for HDL-C, apo-A1, TC, and LDL-C with increase in alcohol intake (lifelong abstainers, ≤12 g/day, 13–24 g/day, 25–47 g/day, 48–96 g/day, and >96 g/day).
**INTERVENTIONALS STUDIES**				
Padro et al. [25]	RCT, *N* = 36	Healthy men and women (40–60 y)	Effect of beer on weight, lipoproteins, and vascular endothelial function	Consumption of beer (2 servings/d for men and 1 serving/d for women) promotes protective properties of HDL-C.
Chiva-Blanch et al. [46]	RCT, *N* = 73	High CV risk, men	Serum lipids	LDL-C decreased 4.5% after gin and red wine (30 g alcohol/d each) consumption, HDL-C increased 5% and 7% respectively.
Mori et al. [62]	RCT, *N* = 24	Healthy premenopausal women	Changes in BP	Higher level of RW consumption (>200 g/week) is associated with increase in HDL-C.
Kechagias et al. [72]	RCT, *N* = 44	Healthy men and women	Effect of wine on hepatic steatosis	Moderate consumption (30 g/day) increased HDL-C by 5% and apo-A-I by 6%.
Chiva-Blanch et al. [73]	RCT, *N* = 33	High-risk men, Spain	CV risk factors	Moderate consumption (30 g/day) increased HDL-C by 5% and apo-A-I by 6%.
**META-ANALYSES**				
Brien et al. [32]	Meta-analysis of 63 interventional studies, *N* = 1686	Adults with known cardiovascular disease	Biological markers of CHD	Dose–response increase in HDL-C and Apo- AI, intake >60 g/day increases TG.

Apo-AI, apolipoprotein AI; CHD, coronary heart disease; CV, cardiovascular; HDL-C, high-density lipoprotein cholesterol; HTG, hypertriglyceridemia; LDL-C, low-density lipoprotein cholesterol; RCT, randomized controlled study; RW: red wine; TC, total cholesterol; TG, triglycerides.

**Table 4 nutrients-12-00912-t004:** Characteristics of eight studies evaluating the effect of alcohol on Type 2 diabetes.

Reference	Design, Subjects (No), Follow-up	Population	Primary Outcome	Main Results
**OBSERVATIONAL STUDIES**				
Da Luz et al. [38]	Cohort study, *N* = 354, 5.5 years	Healthy men, Brazil	Total mortality, AMI, or coronary revascularization	RW consumption (28.9 ± 15 g/d) was associated with better event-free survival
Waśkiewicz et al. [55]	Cross-sectional, *N* = 6912	National Multicenter Health Survey (WOBASZ), male population, Poland	CV risk factors	Moderate intake (15–30 g) is associated with 35% lower risk of T2D
Joosten et al. [75]	Cohort study, *N* = 35,625, 10.3 years	EPIC-NL: Low-risk population (20–70 y), Netherlands	T2D incidence	Moderate intake (5.0–29.9 g/d) reduces risk of T2D in low-risk lifestyle behavior
Joosten et al. [76]	Cohort study, *N* = 38,031, 11.2 years	Health Professionals Follow-Up Study (HPFS). Healthy men, (40–75 y)	T2D incidence	A 7.5 g/day increase in consumption over 4 years is associated with lower T2D risk in non-drinkers and low drinkers (<15 g/d)
Márquez-Vidal et al. [77]	Cohort study, *N* = 4765, 5.5 years	CoLaus study, general population, Lausanne (35–75 y)	T2D incidence	Moderate intake (14–27 units/week) is associated with lower risk of T2D
Koloverou et al. [78]	Cohort study, *N* = 3042, 10 years	ATTICA study, general population, Greece (18–89 y)	T2D incidence	Low intake (1 glass/d) associated with 53% lower T2D risk (RR 0.47), U-shaped relationship between alcohol intake and T2D incidence
Strelitz et al. [79]	Cohort study, *N* = 852, 10 years	ADDITION-Cambridge, T2D patients, England (40–69 y)	Incidence of CV events and all-cause mortality	Decreasing intake by ≥2 units/week is associated with lower HR of CVD.

**META-ANALYSES**				
Baliunas et al. [80]	Meta-analysis of 20 cohort studies, *N* = 477,200	Adults with known CV disease	T2D incidence	Moderate alcohol consumption is protective for T2D in men (22 g/day) and women (24 g/day).

AMI, acute myocardial infarction; CV, cardiovascular; HR, Hazard Ratio; RW, red wine; T2D, Type 2 Diabetes.

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
