# Peer review of "The Effect of Alcohol on Cardiovascular Risk Factors: Is There New Information?"

_nutrients, 2020, doi:10.3390/nu12040912_

Round 1
Reviewer 1 Report
This review article aims to examine the relationship between alcohol consumption and cardiovascular risks and its underlying mechanisms of damage.
However, suggestions should be addressed.
-Each table describes human study design. However, it is hard to clearly see if how many RCTs, observational studies, meta-analyses and so on were mentioned in each table. Therefore, I would suggest that authors had better address this number of designs in texts related all tables.
-Authors should clarify if studies used in meta-analyses articles overlap studies mentioned in this review. So, authors had better to put references of each study of meta-analyses.
For example, if authors mentioned a meta-analysis of 12 prospective studies in texts and/or in tables. 12 prospective studies can be cited.
Author Response
We provide a point-by-point response to the reviewer’s comments

Reviewer 2 Report
This study has summarized the literature on the association between alcohol consumption and cardiovascular risk factors. However, there are somethings that the author did not mention.
- for defining the alcohol consumption, each country uses different way to quantify one standard drink. The author needs to give general/multiple definitions of light-top-moderate drinking and heavy drinking.
- What about the modifying effect of behavioral factors on the link between alcohol consumption and cardiovascular risk factors
- Are the associations differ by population/race/gender/education level?
Author Response

(The authors gave the same response as above.)

Reviewer 3 Report
The topic of the present work is very interesting and this study could give an update of scientific evidence on the effect of alcohol consumption on cardiovascular risk factors. However, there are some methodological problems in the performing this review.
The title of the manuscript suggests a more extensive analysis of the issue on the role of alcohol consumption in etiology of CVD, that has been studied through its relationship with the major cardiovascular risk factors by reporting all recent literature but without following a clear design of the work.
First of all, in the manuscript there isn’t an explanation about how this review was done, please the Authors should provide detailed information how the literature search was performed (when, key words, databases…) and any kind of inclusion and exclusion criteria and quality control for the included papers. The Journal requests that Systematic reviews should follow the PRISMA guidelines.
There isn’t a Discussion section and the conclusions don’t show a final and clear key message.
Additionally, the Authors can find below several comments about their manuscript, that, in my opinion, they could be useful to revise and improve the full paper.
Introduction:
- Page 1/24, line 27. Please add reference for the first sentence.
- I suggest to delete figure 1, no comments or discussion were made in the text about that. The figure is too much “minimalist”. However, if the Authors decide to maintain the figure 1, there is a typo in the second box: “pattern ok intake” (replace ok with of ) and I suggest to add in the box “gender, quantity, pattern...” also age and individual health profile.
- Page 2/24, lines 37-38: in my opinion, <<Brief Intervention: For Hazardous and Harmful Drinking. http://whqlibdoc.who.int/hq/2001/WHO_MSD_MSB_01.6b.pdf?ua=1%20>> is the correct reference for the following sentence: “A standard drink has different definitions depending on the country and guideline revised. According to the World Health Organization (WHO) it contains 10 grams of pure alcohol”.
- I suggest to delete the following sentence “It can also be quantified in units, as 1 unit of alcohol equals 10 mL or 8 g of ethanol, in reference to the amount an adult can metabolize in one hour [3,4]”, because it is confusing to the reader. The Authors should only use “standard drink” to quantify the alcohol consumption.
- Line 82: please define “major adverse cardiovascular event”
- The findings of study #25 were reported in not complete manner: “They described that abstinent participants (zero drinking days/year) and those who drank frequently (>100 drinking days/year) exhibited a higher NLR (2.06 and 2.01, respectively) than less- frequent drinkers [25].” Please add value of NLR for less-frequent drinkers.
Chapter 2: Pathophysiology and Oxidative Stress
In the all chapters, the “Main results” from the selected articles have been unclearly reported, as well in the Tables. Below, I’m reporting some examples:
- In the table 1 the Authors wrote for study #31: “Alcohol consumption may have beneficial effects on inflammation and hemostasis in postmenopausal women “. Any dose or pattern of alcohol have beneficial effects? Be careful to explain and report results from selected studies. Nowadays, scientists are not interested if drinking alcohol (yes or no) has beneficial or harmful effect on heath, but the real question is how much alcohol an individual can drink to get health benefits and viceversa.
- Table 1, Column “Design, Cases (No), Follow-up”:
- replace Cases with Subjects;
- for example, the study #27 is not a cohort study. please check the type of study design of all quoted studies;
- Study #34: “Mendelian randomization cross Sectional study (confounder-adjusted multivariable and IV analyses)” this statement is unclear;
- I suggest to separate the table1 in 2 sections: interventional studies and observational studies.
- In general, when the Authors report findings from other studies they should consider also the definition of alcohol drinking categories used in the selected studies.
- For example study #46 “concluded that moderate alcohol drinkers had 37% higher risk of hypertension while heavy drinkers had 52% more risk [46]. ” Checking in the method section of study #46, about the alcohol categorization, Waśkiewicz et al specified : “The studied subjects were divided into 4 groups: abstainers (A), light drinkers (L; ≤ 15 g ethanol/day), moderate drinkers (M; 15-30 g ethanol/day), and heavy drinkers (H; > 30 g ethanol/day).” The Authors can see that moderate Polish drinkers consumed more than one standard drink per day. Additionally, the referent category used in this study was light drinkers and not abstainers, see careful table 3 of study #46. Please, take in to account the importance of the selection of the reference group and how the use of light-moderate drinkers as reference group exacerbates the risk for moderate and higher intake categories in this study.
- Chapters 4 and 5. Please add a brief definition of Dyslipidemia and Type 2 Diabetes Mellitus, respectively.
- lines 377-381. Please move the paragraph on #65 “A light-to-moderate alcohol consumption has been associated with a lower risk of Type 2 Diabetes Mellitus (T2D) in non-high-risk volunteers. Yet, in recently diagnosed patients, small reductions in alcohol use are associated with lower hazard of CVD. A potential positive effect of moderate alcohol consumption, as manifested by higher HDL-C level, could be counterbalanced by a negative effect on blood pressure in high-risk patients [65].” before the reported findings from #70.
- Brief Conclusions statement in the chapters:
- At the end of chapter 2 the Authors reported only conclusions on Anti-inflammatory and antioxidant effects and anything else on Anti-thrombotic effects
- There isn’t Conclusions statement in chapter 4 on dyslipidemia, as well as in chapter 5 on T2D.
- About tables’ Title:
- Table 3. Characteristics of 14 studies evaluating alcohol’s effect on Dyslipidemia. In the table 3, 13 studies have been described
- Table 4. Characteristics of 14 studies evaluating alcohol’s effect on Type 2 Diabetes. In the table4, 7 studies have been described
Author Response

(The authors gave the same response as above.)

Round 2
Reviewer 1 Report
This manuscript has been revised based on comments.
It is acceptable to be published.
Reviewer 3 Report
The revised manuscript is well written and provides a rationale interpretation of the existing scientific evidence on the effect of alcohol consumption on cardiovascular risk factors